# Smartphone-Based Deep Learning System for Detecting Ractopamine-Fed Pork Using Visual Classification Techniques

**DOI:** 10.3390/s25092698

**Published:** 2025-04-24

**Authors:** Hong-Dar Lin, Mao-Quan He, Chou-Hsien Lin

**Affiliations:** 1Department of Industrial Engineering and Management, Chaoyang University of Technology, Taichung 413310, Taiwan; s10915604@gm.cyut.edu.tw; 2Department of Civil, Architectural, and Environmental Engineering, The University of Texas at Austin, Austin, TX 78712-0273, USA; chslin@utexas.edu

**Keywords:** ractopamine-fed pork, visual inspection, computer vision, deep learning, MobileNet

## Abstract

Ractopamine, a beta-agonist used to enhance lean meat yield, poses health risks with excessive consumption. To comply with global trade policies, Taiwan permits imports of North American (USA. and Canadian) pork containing ractopamine, raising concerns over unclear labeling and potential misidentification as Taiwan pork. Given the high demand for pork, consumers need a reliable way to verify meat authenticity. To address this issue, this study proposes a smartphone-based visual detection system for meat cut and pork origin classification, extending to ractopamine detection. Consumers can use mobile devices in retail settings to analyze pork images and make informed decisions. The system employs a three-stage process: first, applying a black elliptical mask to extract the outer ROI (region of interest) for meat cut classification; then, using a black square mask to obtain the inner ROI for pork origin classification; and finally, determining ractopamine presence in North American pork. Experimental results demonstrate MobileNet’s superior accuracy and efficiency, achieving a 96% CR (classification rate) for meat cut classification, a 79.11% average CR and 90.25% F1 score for pork origin classification, and an 80.67% average CR and 80.56% F1 score for ractopamine detection. These findings confirm the system’s effectiveness in enhancing meat authenticity verification and market transparency.

## 1. Introduction and Background

Beta-adrenergic agonists, such as ractopamine, are used in some countries’ livestock industries to enhance profitability. Developed specifically as a growth promoter, ractopamine increases lean muscle mass, improves feed efficiency, and reduces production costs in pigs and cattle. Pigs fed ractopamine often exhibit well-defined hind muscles and a leaner appearance, making them more marketable. After slaughter, the meat from these animals is typically leaner, although studies suggest it may not be significantly redder and could appear slightly paler due to changes in muscle pH [1]. This lean quality can appeal to consumers seeking healthier-looking pork.

Beta-adrenergic agonists vary in composition, affecting their metabolic rates and pharmacological effects in animals. Some countries, like the U.S. and Canada, allow ractopamine in animal feed, while others restrict it to medical use. The Codex Alimentarius Commission (CAC) has set maximum residue limits (MRLs) for these substances in food. On August 28, 2020, Taiwan announced that starting in 2021, it would permit the import of U.S. pork containing ractopamine and U.S. beef from cattle over 30 months old, sparking public controversy [2].

The maximum residue limits (MRLs) for ractopamine have been widely debated, and it was not until 5 July 2012, that the Codex Alimentarius Commission (CODEX) established an official MRL standard. Taiwan does not restrict specific pork cuts for import but enforces ractopamine residue limits: 0.01 ppm for muscle and fat (including skin), 0.04 ppm for liver and kidneys, and 0.01 ppm for other edible parts (e.g., stomach, intestines, heart, lungs, tongue, tripe, brain, and blood). The higher limit for the liver and kidneys reflects their detoxification function [2].

Taiwan’s self-sufficiency in pork exceeds 90%, with imports making up less than 10%. The majority of imported pork comes from Canada (37%) and Spain (19%), while U.S. pork accounts for 16% [3]. Only 22% of North American pork contains ractopamine, while European pork does not. To protect domestic pork competitiveness, Taiwan has banned ractopamine in all domestic pork and will not approve ractopamine-based products [2]. The government requires clear labeling of pork origin in supermarkets, markets, and restaurants, allowing consumers to distinguish between domestic and imported pork.

### 1.1. Pork Appearance Characteristics

A whole pig is divided into five main sections: shoulder, loin, belly, small tenderloin, and hind leg. The upper shoulder is called the pork shoulder, and the lower part is the heart of the shoulder. The central loin section is the tenderloin, while the small tenderloin is a lean cut beneath the backbone, connected to the rib. The belly section contains pork belly, and the upper hind leg is the ham. Table 1 compares their classification and characteristics [4].

Table 2 compares Taiwan pork and U.S. ractopamine-fed pork, highlighting differences in color, texture, fat distribution, and structure. The lighting conditions are standardized during data collection using 5000 K daylight-balanced LED lighting at an intensity of 1000 lumens, ensuring consistent illumination across the two sample sets. Pork without ractopamine typically has a thick fat layer beneath the skin, while ractopamine-fed pork has a thin fat layer or skin attached to the lean meat. Normal pork has light pink lean meat and white fat, whereas ractopamine-fed pork has brighter red lean meat and less white fat. For pork belly, the subcutaneous fat should be at least 1 cm thick. Visually distinguishing between these types of pork can be challenging.

### 1.2. Research Motivation and Purpose

Pork is a widely consumed staple, but food fraud, such as mislabeling its origin, threatens consumer trust and market transparency. Ractopamine enhances pork’s visual appeal, making it easier to mislabel lower-cost North American pork as premium Taiwanese pork. Although regulations mandate clear origin labeling, varied packaging and products often result in unclear or misleading labels, undermining consumer confidence in food safety and quality.

Although food fraud rarely poses immediate health risks, it damages public trust in food systems. Inconsistent labeling misleads consumers, who depend on accurate information for informed choices. Existing methods for verifying pork authenticity are impractical at the point of sale, highlighting the need for a simple, real-time tool to detect ractopamine residues and confirm pork origin, promoting transparency and protecting consumer rights.

This study introduces a smartphone-based visual detection system for identifying ractopamine residues in North American pork before purchase. Using deep learning and cloud-based image analysis, the system extracts visual features from segmented meat images to classify pork in real time. It addresses the challenges of varied cuts and packaging, offering a fast, user-friendly tool for verifying pork authenticity. By enabling real-time detection and origin verification, the system helps prevent meat fraud, supports informed purchasing, and strengthens consumer trust and supply chain transparency.

This article first reviews existing ractopamine inspection methods and then introduces the proposed three-stage deep learning models for pork classification and ractopamine detection. It next presents performance evaluations and robustness analysis, followed by a summary of key contributions and future research directions.

## 2. Review of Prior and Related Work

### 2.1. Pork Quality and Inspection

Pork quality varies by meat cut due to the fat-to-lean ratio, influencing taste and cooking suitability. Freshness also affects quality and is typically evaluated based on color, texture, and marbling. Various methods have been developed for assessment. Barbin et al. [5] used hyperspectral imaging with PCA to classify pork into three quality grades. Sun et al. [6] applied an SVM-based computer vision system to predict color and marbling from loin images. Liu et al. [7] showed that both stepwise regression and SVM models effectively estimated intramuscular fat percentage (IMF%) using image analysis.

Recent advancements in imaging technologies and machine learning have significantly improved pork quality assessment. Zhuang et al. [8] demonstrated the use of fluorescence hyperspectral imaging to assess frozen pork freshness without thawing. Huang et al. [9] predicted IMF content and marbling scores using texture features and regression analysis. Ma et al. [10] showed that Vis–NIR imaging combined with spectral preprocessing accurately estimates IMF in cooked pork loin. Zheng et al. [11] integrated thermal imaging with CNNs to detect pork adulteration in minced lamb, achieving high accuracy, while Sun et al. [12] used hyperspectral backscattering imaging with an LD model for non-destructive tenderness evaluation. Together, these studies highlight the potential of combining advanced imaging and AI techniques for rapid, accurate, and non-invasive pork quality assessment.

### 2.2. Testing Methods for Pork Containing Ractopamine

Ractopamine and other β-agonists in meat are typically detected using laboratory-based methods involving hydrolysis, extraction, purification, and LC-MS/MS (liquid chromatography-tandem mass spectrometry) analysis, offering high accuracy and broad detection capabilities. Dong et al. [13] developed a rapid fluorescence polarization immunoassay (FPIA) using fluorescein-labeled ractopamine derivatives, demonstrating fast, accurate, and sensitive screening in pork. Yan et al. [14] introduced a simple HPLC-UV method for quantifying salbutamol, ractopamine, and clenbuterol in pork. Chang et al. [15] used LC-MS/MS to detect ractopamine and salbutamol in pig hair, showing potential for feed monitoring. Feddern et al. [16] applied LC-MS to pig organs, confirming its effectiveness across various tissue types.

Chen et al. [17] used isothermal titration calorimetry to identify an AP-Ago activator and developed a label-free electrochemical sensor with a gold electrode for β-agonist detection, showing high reproducibility and rapid screening capability. Yin et al. [18] employed UPLC-MS/MS with positive ESI and scheduled MRM mode to detect 210 drug residues in pork within 20 min, offering a fast, cost-effective, and comprehensive screening method. From the above literature, most β-agonist detection methods rely on chemical instruments and have shown promising results.

Yin et al. [19] utilized Raman spectroscopy and a deep learning model to predict ractopamine concentrations in pork. Spectral data, visualized as graphs or chemical maps, capture molecular vibrations and chemical compositions (e.g., ractopamine, proteins). Raman images depict chemical distribution via molecular signals, not visible light. The costly system requires controlled lab conditions to minimize interference from ambient light or fluorescence.

### 2.3. Analysis of the Texture Characteristics of Pork

Pork quality is often reflected in its appearance, including color, marbling, and texture. Sun et al. [20] used image processing and regression to predict pork loin color based on 18 features from RGB, HSI, and Lab* spaces, highlighting computer vision’s potential. Uttaro et al. [21] applied image analysis with cyan-filtered monochrome JPEGs to detect marbling in boneless pork loin. Huang et al. [22] developed a fast NIR hyperspectral method for assessing IMF in pork ribs. Chmiel et al. [23] demonstrated PSE defect detection using V/B (HSV/HSB), L (HSL), and RGB values in color lightness of pork. Huang et al. [24] also used hyperspectral imaging to assess marbling in the longissimus thoracis muscle with strong predictive accuracy. These studies confirm the effectiveness of image processing in extracting color and texture features for marbling analysis.

### 2.4. Deep Learning Models Based on Neural Network Frameworks

In recent years, deep learning has seen the fastest growth in classification technology, driven by continuous improvements in GPU hardware and software, which have significantly enhanced computational power. Many deep learning models are now applied across various fields, including LeNet, one of the earliest CNNs; AlexNet, which had a major impact on computer vision; VGG16, which extracts more features by deepening the network; InceptionV3, which utilizes multiple small convolutional modules to enhance depth; Xception, which reduces the number of parameters to speed up training; Custom Vision, which combines pre-trained CNNs with transfer learning to increase efficiency in model training; and MobileNet, designed for mobile devices.

VGG (Visual Geometry Group) is a deep CNN known for its simple yet effective architecture, using uniform layers and small 3×3 filters. It performs well in image classification, object detection, and feature extraction, making it widely used for transfer learning across domains such as medical imaging, industrial inspection, and agriculture. Applications include mango farm threat detection [25], smartphone-based eggplant disease recognition [26], outdoor fruit detection [27], and date fruit classification [28]. VGG-19 achieved top accuracy in papaya ripeness classification [29], while VGG-16 has been integrated into a kiwi-picking robot [30] and used for fish freshness detection with high accuracy [31].

InceptionV3 is a deep CNN that captures multi-scale features through parallel convolutions, using 1 × 1 filters to reduce computational cost. It is both efficient and accurate and widely applied in fields like autonomous driving and medical imaging. Sathish et al. [32] achieved 95.30% accuracy in Indian dish classification and calorie estimation, while Ma et al. [33,34] reported 94% and 96.62% accuracy with Chinese food datasets. The model has also been successfully applied to weed detection [35], pest-infected maize leaf classification [36], and grouper fish abnormality detection [37].

Xception (Extreme Inception) enhances the Inception architecture using depthwise separable convolutions, reducing complexity while maintaining high accuracy. Known for its efficiency and strong performance, it is widely used in image classification, object detection, medical imaging, and autonomous driving. Xception has been applied to wood species classification with high accuracy [38], sorghum classification [39], and apple disease detection using Faster R-CNN, achieving 88% accuracy [40]. Grijalva et al. [41] also used Xception and InceptionV3 to classify sugarcane aphid density with 86% accuracy, supporting pest management efforts.

MobileNet is a lightweight neural network optimized for mobile and embedded systems, using depthwise separable convolutions to reduce computation while maintaining accuracy. Its adjustable width and resolution multipliers make it well-suited for real-time tasks in image recognition, autonomous driving, and medical imaging. Kc et al. [42] achieved high accuracy in plant disease detection with Reduced MobileNet, while Ashwinkumar et al. [43] surpassed state-of-the-art methods in plant classification. Prasetyo et al. [44] used a revised version for fish eye freshness classification with moderate accuracy. Enhanced versions, such as MobileNet-V2 with attention [45] and MobileNet with Rmsprop [46], showed high accuracy in rice and olive disease detection, respectively. Xiao et al. [47] developed a lightweight MobileNet model for orchid growth stage classification, achieving high accuracy. These demonstrate its suitability for mobile vision applications.

Microsoft Custom Vision [48] is a cloud-based service that streamlines image classification and object detection using deep learning, transfer learning, and CNNs with minimal coding. With automated hyperparameter tuning and integration with Azure Machine Learning, it enables scalable, real-time deployment using API [49,50]. This makes it a practical tool for applications such as pork cut recognition, origin identification, and ractopamine detection.

### 2.5. Research Gap and Limitations

The literature above demonstrates that many studies have applied computer vision technology to various aspects of pork inspection, including analyzing IMF content [7,10], marbling [6], freshness [8], tenderness [12], and quality grading [5]. However, none of the reviewed studies mention the use of techniques for processing and analyzing visual images to detect whether pork has been produced using ractopamine, a concern of significant importance to consumers. To underscore the current gap in ractopamine detection, we emphasize the lack of consumer-friendly, smartphone-based tools, despite the availability of advanced laboratory methods. Techniques such as LC-MS/MS [13,14,15,16,18] and Raman spectroscopy [19] provide high accuracy but remain inaccessible to most consumers due to their cost, complexity, and long processing times. Our study addresses this gap by introducing a smartphone-compatible, deep learning–based system that utilizes cloud-transmitted visual classification to detect ractopamine in pork under various conditions, including fresh, refrigerated, and post-defrost refrigerated samples. This real-time, user-friendly solution empowers non-experts to make informed purchasing decisions, enhancing food transparency and consumer confidence.

## 3. Proposed Methods

This study proposes a mobile-based system that enables consumers to capture and analyze pork images in retail settings to detect ractopamine before purchase. The process involves three stages: (1) classifying meat cuts using an outer ROI (region of interest) with a black elliptical mask, (2) determining pork origin using an inner ROI with a black square mask, and (3) detecting ractopamine in pork from the U.S. and Canada.

The system development process is divided into four steps. The first step involves capturing test images using a handheld mobile device and extracting the outer and inner ROI areas with dual-layer masks. The second step involves creating deep learning models for meat cut classification, origin classification, and ractopamine detection; building training and testing datasets; adjusting parameters; and training using the training dataset. The third step tests the models with the testing dataset; analyzes the classification results of pork cuts, origin, and ractopamine detection; and creates a confusion matrix for evaluation. The fourth step involves evaluating and comparing the model’s effectiveness with other deep learning models and outputs the classification and detection results.

### 3.1. Image Capture

This study aims to empower consumers to capture images of fresh, refrigerated, and post-defrost refrigerated pork in retail settings using handheld mobile devices, enabling on-site analysis to determine the presence of ractopamine before purchase. Figure 1 illustrates the methods for capturing images of pork. Vertical capture involves holding the mobile device horizontally and shooting from above. The 45° angle capture is taken from an approximately 45° angle. The horizontal capture is taken from the side of the pork.

### 3.2. Extraction of the Outer and Inner ROIs in a Pork Image

To minimize variations in image acquisition caused by background noise or labels in a retail environment, this study applies a black elliptical mask to extract the outer ROI after capturing images. The outer ROI area is 305,363 pixels, as shown in Figure 2. Assuming the original image M(X, Y) in this study has a size of 960 × 720 pixels, there may be unwanted objects in the image that could interfere with the analysis. To eliminate these distractions, a black elliptical mask, as shown in Figure 2b, is applied. The center of the ellipse is set at the midpoint of the image dimensions, M(480,360). The semi-major axis extends 360 pixels along the X-axis in both directions, while the semi-minor axis extends 270 pixels along the Y-axis in both directions. Figure 2c shows the image after applying the elliptical mask. In this study, the masked image with the extracted outer ROI is used as the input for meat cut classification.

After extracting the outer ROI using a black elliptical mask, this study further applies a black square mask to extract the inner ROI, which has an area of 50,176 pixels, as shown in Figure 3. Since unwanted objects in the test image can negatively impact classification performance, the black square mask is used to exclude these interferences, as illustrated in Figure 3b. The square mask has a width and height of 224 pixels, expanding 112 pixels along both the X- and Y-axes from the image center point M(480,360). Figure 3c shows the image after applying the elliptical mask. The masked image, with the extracted inner ROI, is then used as input for pork origin classification and ractopamine detection.

In this study, two different-sized ROIs are used to enhance the accuracy of pork classification and ractopamine detection by focusing on distinct visual features. The larger outer ROI, extracted using a black elliptical mask, is designed for meat cut classification, as it captures the overall shape, texture, and fat distribution of the pork, which are essential for distinguishing different cuts. In contrast, the smaller inner ROI, obtained through a black square mask, is optimized for pork origin classification and ractopamine detection, as it isolates the central portion of the meat, minimizing external interference and focusing on finer details such as marbling, color, and texture variations. This two-stage ROI extraction method ensures that each classification task is performed on the most relevant region of the image, improving both efficiency and accuracy while reducing the risk of misclassification due to background noise or non-essential features.

### 3.3. Deep Learning Models Applied to Pork Image Classification and Ractopamine Detection

After applying black elliptical and square masks to obtain the outer and inner ROI regions, this study establishes deep learning models and training/testing datasets for meat cut classification, pork origin classification, and ractopamine detection in North American pork. The classification tree is shown in Figure 4. The classification and detection process then utilizes deep learning models. Here, pork is first classified into five meat cuts; followed by origin classification into three categories; and finally, North American pork is analyzed for ractopamine presence.

#### 3.3.1. Deep Learning Network Models

In this study, five deep learning models, including VGG16, InceptionV3, Xception, MobileNet, and Microsoft Custom Vision, are employed due to their shared foundational characteristics suited for image-based pork classification and ractopamine detection. All models are based on CNN architectures, enabling them to learn hierarchical spatial features critical for distinguishing subtle visual differences in meat texture, color, and surface characteristics. They are initialized with ImageNet-pretrained weights, utilizing transfer learning to improve generalization and reduce the data requirements specific to pork imagery. These models are applied both as feature extractors and as fully fine-tuned networks, depending on the experimental setup. Each network supports image classification tasks and generates probabilistic outputs suitable for both binary scenarios (e.g., presence or absence of ractopamine) and multi-class scenarios (e.g., different meat cuts).

#### 3.3.2. Parameter Settings for Deep Learning Network Models

This study utilizes the MobileNet architecture and other deep learning models for classification and detection, followed by performance evaluation. In deep learning network models, several parameters must be adjusted, including the learning rate, batch size, optimizer, and number of epochs. This study conducts two types of parameter-setting experiments based on different image sizes. Table 3 presents the default parameter settings for various network models with different image sizes. A detailed explanation of each parameter and its impact on classification results can be found in the following sections.

#### 3.3.3. MobileNet Model Applied to Pork Classification and Ractopamine Detection

A three-stage sequential classification network using the MobileNet model consists of three separate MobileNet classifiers, each responsible for progressively refining the classification process. The first stage, meat cuts classification, determines the type of pork cut (e.g., shoulder, loin, belly, ham, tenderloin). This helps narrow down the search space for later classification stages. Based on the identified meat cut, the second stage, pork origin classification, classifies the country of origin (e.g., Taiwan, Europe, USA/Canada). This helps in identifying whether the pork is from a region that might use ractopamine. The final stage, ractopamine detection, determines whether the pork from the USA/Canada contains ractopamine or not. Each stage uses MobileNet as the backbone deep learning model, which is lightweight and efficient, making it suitable for smartphone-based applications.

Each stage in the three-stage sequential classification network follows a MobileNet-based architecture, fine-tuned for its specific classification task. The input layer processes RGB images with a resolution of 960 × 720 × 3 for meat cut classification and 224 × 224 × 3 for pork origin and ractopamine detection. Feature extraction is performed using depthwise separable convolutions, reducing computational complexity through Conv-BN-ReLU blocks, starting with a standard convolution layer (stride = 2), followed by 13 depthwise separable convolution layers, each with batch normalization (BN) and ReLU activation. This structure progressively reduces spatial resolution while enhancing feature depth. A global average pooling (GAP) layer then condenses spatial dimensions into a single vector, retaining essential feature representations. Finally, a fully connected (FC) layer maps extracted features to class probabilities, employing Softmax activation for multi-class classification.

In this study, after extracting the outer and inner ROI regions of pork images using a dual-layer ROI approach, the outer ROI region, obtained through a black elliptical mask, is classified into different pork cuts using the MobileNet deep learning model. The training process is illustrated in Figure 5. First, the extracted outer ROI region is input into the model with an image size of 960 × 720, which is resized to 224 × 224 using the first convolutional layer. MobileNet applies depthwise separable convolutions by splitting a standard convolution into two steps: depthwise convolution, where each filter processes a single input channel, and pointwise convolution, which combines the output. This reduces computation and the number of parameters while maintaining good performance. Finally, the fully connected layer and Softmax activation function calculate probabilities to classify the pork cuts into five categories. 

After classifying the pork cuts, this study extracts the inner ROI region using a black square mask. Based on the classification results of the five pork cuts, the model further classifies the pork into three different origins. Finally, it determines whether North American pork contains ractopamine. Figure 6 and Figure 7 illustrate the training process using the MobileNet model for pork origin classification and hormone detection. In addition to MobileNet, this study also evaluates other deep learning models for comparison.

#### 3.3.4. Sequential Classification and Multi-Class Classification Approaches

In the context of pork classification and ractopamine detection, sequential classification is advantageous because it allows a step-by-step refinement, from meat cut identification to origin classification and finally to ractopamine detection. This method reduces the number of categories at each stage, improving classification accuracy, especially when certain stages influence subsequent predictions (e.g., only pork from certain regions requires ractopamine detection). However, if computational efficiency is a priority and all categories are independent, a multi-class classification approach might be preferred, as it simplifies the process by training a single model to handle all classifications at once.

Figure 8 compares the overall sequential classification and multi-class classification. In sequential classification, the process is divided into three hierarchical stages: meat cuts classification (Model 1), pork origin classification (Model 2), and ractopamine detection (Model 3), where each stage refines the classification further based on the previous one. In contrast, the multi-class classification approach uses a single model to classify the input directly into one of 20 predefined categories, combining all classification tasks into one step. The sequential approach provides a structured decision-making process, while the multi-class method simplifies classification by handling all categories simultaneously.

Figure 9 illustrates a three-stage smartphone-based system for pork image analysis using deep learning. Image preprocessing extracts two regions of interest (ROIs) from a captured pork image. Stage 1 classifies the meat cut into one of five categories. Stage 2 determines the pork’s origin among Taiwan, North America, and Europe. Stage 3 detects the presence or absence of ractopamine. Each stage uses a CNN model for visual classification, enabling real-time, non-invasive assessment of pork type, origin, and safety.

## 4. Experiments and Results

This study utilizes a personal computer with the following specifications: AMD Ryzen^TM^ 9 5900HS CPU, 48 GB RAM, GeForce^®^ RTX 3060 Laptop GPU, and Windows 10 operating system. The pork classification and ractopamine detection system is developed using Python (version 3.8.5). Initially, meat cut images are resized to 960 × 720 pixels, while images for pork origin classification and hormone detection are resized to 224 × 224 pixels. The system is implemented using TensorFlow and Keras libraries to build a CNN model. Various parameters are adjusted to optimize performance, and the final results include pork classification accuracy and the detection of ractopamine. A user interface prototype of the system is shown in Figure 10. In the images of this figure, the red ellipse is the outer ROI for classifying meat cuts, the blue square is the inner ROI for determining origin, and the yellow square is the inner ROI for detecting ractopamine. 

### 4.1. Performance Evaluation Metrics for the Detection System

After conducting the detection process, the system’s effectiveness is assessed by determining whether the pork samples are correctly classified. To validate the classification performance, a confusion matrix is used to present classification outcomes. The three-level evaluations are based on recall, precision, and classification rate (CR, accuracy). Additionally, the F1 score is derived from recall and precision to provide a more comprehensive assessment of the proposed method’s performance.

1.Effectiveness metric for meat cut classification

The classification rate (*CR*%) for meat cut classification is defined as follows:(1)$$Meat\;cut\;(CR\%)=\left(\frac{\text{Number of correctly classified images for each meat cut category}}{\text{Total number of test images}}\right)\times 100\%$$

2.Effectiveness metrics for pork origin classification

Recall (*R*%) for North American pork is defined as follows:(2)$$Pork\;origin\;(R\%)=\left(\frac{\text{Number of correctly classified North American pork images}}{\text{Total number of actual North American pork images}}\right)\times 100\%$$

Precision (*P*%) for North American pork is defined as follows:(3)$$Pork\;origin\;(P\%)=\left(\frac{\text{Number of correctly classified North American pork images}}{\text{Total number of detected North American pork images}}\right)\times 100\%$$

The classification rate (*CR*%) for pork origin is defined as follows:(4)$$Pork\;origin\;(CR\%)=\left(\frac{\text{Number of correctly classified pork origin images}}{\text{Total number of test images}}\right)\times 100\%$$

3.Effectiveness metrics for detecting ractopamine contained in North American pork

Recall (*R*%) for ractopamine-containing pork is defined as follows:(5)$$Ractopamine\;(R\%)=\left(\frac{\text{Number of correctly detected ractopamine containing North American pork images}}{\text{Total number of actual ractopamine containing North American pork images}}\right)\times 100\%$$

Precision (*P*%) for ractopamine-containing pork is defined as follows:(6)$$Ractopamine\;(P\%)=\left(\frac{\text{Number of correctly detected ractopamine containing North American pork images}}{\text{Total number of detected ractopamine containing North American pork images}}\right)\times 100\%$$

The classification rate (*CR*%) for ractopamine detection in North American pork is defined as follows:(7)$$Ractopamine\left(CR\%\right)=\left(\frac{\text{Number of correctly classified ractopamine containing or free North American pork images}}{\text{Total number of test images}}\right)\times 100\%$$

The F1 score (%) for North American pork or ractopamine-containing pork is defined as follows:(8)$$F1\;Score\%=\left(\frac{2\times Recall\times Precision}{Recall+Precision}\right)\times 100\%$$

### 4.2. Parameter Settings of Deep Learning Models for Image Classification

At this stage of the study, parameter adjustments are conducted for both pork cut classification and pork origin classification, as well as ractopamine detection, to enhance the robustness of the network model. The pork cut classification images have a resolution of 960 × 720 pixels, with 150 images (100 for training, 50 for testing). The pork origin classification and ractopamine detection images are 224 × 224 pixels, with 450 images for origin classification (300 for training, 150 for testing) and 300 images for ractopamine detection (200 for training, 100 for testing), as shown in Table 4. The training and testing images are selected using a 2:1 split, with stratified sampling applied to ensure a balanced representation of classes (e.g., equal numbers of ractopamine-positive and ractopamine-negative samples). Class balancing is achieved by ensuring that each class has an equal number of images within its respective stage. This approach minimizes bias toward any particular class and supports robust model training and evaluation. During the training process, the training dataset is further divided into 90% for training and 10% for validation, with the validation set kept separate to ensure an unbiased evaluation of model performance.

The ground-truth labels in this study are determined based on the information provided on the sample packaging, which clearly indicates the treatment conditions. To ensure accuracy, these labels are confirmed using a dual-observer validation process, in which two independent researchers reviewed and verified each label based on experimental records and sample characteristics. Any uncertainties are resolved through discussion to maintain consistency and reliability in the dataset.

MobileNet is selected as the network model for parameter tuning in both cases, with default parameter values set as follows: learning rate, 0.0001; batch size, 16; optimizer, Adam; and training epochs, 20. Various parameter combinations, including different learning rates, batch sizes, optimizers, and training epochs, are tested and compared with the default values to determine the optimal settings. The optimized parameters for pork cut classification and pork origin classification, as well as ractopamine detection, are presented in Table 5 based on comparisons using different parameter combinations and small-sample testing.

### 4.3. Comparisons and Performance Evaluation of Models for Image Classification

After identifying the optimal parameters for the proposed detection method through small-sample experiments, this section conducts experiments using a large sample set. In the large-sample experiment, each category contains three times more images than in the small-sample experiment, while other parameters (e.g., image size and train-test split ratio) remain the same, as shown in Table 6. Since the image sizes differ between meat cut classification and pork origin classification/ractopamine detection, the subsequent performance evaluation will be conducted in two separate phases.

All five models (VGG16, InceptionV3, Xception, MobileNet, and Microsoft Custom Vision) use pre-trained architectures, originally trained on large datasets such as ImageNet, as their basis. This study fine-tunes these models using our labeled pork images and utilizes transfer learning to improve performance on pork classification and ractopamine detection tasks. Additionally, all models require standard image preprocessing, including resizing (e.g., 960 × 720 or 224 × 224), normalization, and data augmentation, to improve robustness.

#### 4.3.1. Meat Cut Classification

This study first classifies five types of meat cuts. Figure 11 presents the effectiveness comparison of large-sample results for meat cut classification and the corresponding line chart of classification rates. The results show that MobileNet achieved the highest classification rate (CR) at 96%, followed by Xception and Custom Vision at 93.33%. Table 7 and Figure 12 compare the execution times of different network models for meat cut classification, revealing that VGG16 and MobileNet required the shortest processing times, while Custom Vision took the longest. These findings indicate that MobileNet is the most suitable choice for meat cut classification in this study.

#### 4.3.2. Pork Origin Classification

Following the classification of five meat cut types, the study continues with the classification of three pork origins. Table 8 and Figure 13 present the performance comparison of different network models for pork origin classification, along with a line chart of performance metrics. As shown in Figure 13, the MobileNet network model outperforms Custom Vision in the average results for pork origin classification. MobileNet achieves an average classification accuracy of 79.11%, followed by Custom Vision at 75.56%. Table 9 and Figure 14 compare the execution times of different network models for pork origin classification. The results show that MobileNet has the shortest processing time, while Custom Vision took the longest. These findings indicate that MobileNet is the most suitable choice for pork origin classification in this study.

#### 4.3.3. Pork Ractopamine Detection

After confirming the origin of each pork cut, this study further examines whether North American pork contains ractopamine. Table 10 and Figure 15 present the performance comparison of different network models for detecting ractopamine, along with a line chart of performance metrics. As shown in Figure 15, the MobileNet network model achieves the best results for classifying the presence of ractopamine, with an average classification accuracy of 80.67%, followed by Custom Vision at 75.33%. Table 11 and Figure 16 compare the execution times of different models for ractopamine detection, showing that both Xception and MobileNet require only 0.03 s per image, while Custom Vision took significantly longer than other models. These results indicate that MobileNet is the most suitable choice for detecting ractopamine in North American pork in this study.

#### 4.3.4. Comparison and Performance Evaluation of Sequential Classification and Multi-Class Classification in the Pork Classification and Ractopamine Detection System

This study employs three-stage classifiers for pork classification and ractopamine detection, aiming to provide users with information on whether the captured pork contains ractopamine for further verification. If only a single result is provided, the overall accuracy of sequential classification can be calculated based on the classification order of the three stages, as shown in Figure 8a. This section explores the effectiveness of integrating all categories into a single system, treating pork classification and ractopamine detection as a multi-class classification approach, as illustrated in Figure 8b, to assess whether this method yields better classification accuracy. Experiments are conducted using both the MobileNet model and Custom Vision to compare the accuracy differences between sequential classification and multi-class classification.

Table 12 shows the comparisons of performance metrics (classification rate and testing time/image) for various classification methods in the pork classification and ractopamine detection system. Using the MobileNet network model, the overall sequential classification accuracy is determined to be 61.27% (0.058 s) when treating the three individual classification accuracies as independent calculations. Similarly, for Custom Vision, the overall sequential classification accuracy is 53.12% (0.764 s). From Table 12, it can be seen that among the three classification stages, pork origin is the most challenging to classify, with an accuracy rate of only 79.11%, while pork cuts are the easiest to classify, achieving an accuracy rate of 96%. When using a single classifier for multi-class classification of the 20 categories, the accuracy rate of the MobileNet model is 60.06% (0.046 s), while that of Custom Vision is 61.8% (0.582 s), showing similar results between the two methods. For classification using the MobileNet model, the accuracy rates of both sequential and multi-class classifications are quite close. If the user provides additional correct information, classification accuracy can be improved. Specifically, if the user clearly identifies the pork cut and its origin, the accuracy of determining whether North American pork contains ractopamine reaches 80.67%. However, if the user is uncertain about the pork cut and its origin, the overall accuracy of the proposed three-stage sequential classification method (61.27%) is slightly higher than that of multi-class classification (60.06%). However, in terms of efficiency, the testing time per image is shorter for multi-class classification (0.046 s) compared to sequential classification (0.058 s).

In the three-stage sequential classification proposed in this study, users can provide additional information during the classification process. This means the method offers greater flexibility, allowing users to obtain more precise detection results. In contrast, the multi-class classification method can only analyze the images captured by users without incorporating additional input. However, when users provide accurate supplementary information, both methods achieve better classification performance. From an efficiency perspective, sequential classification requires training in three stages, resulting in a longer processing time. On the other hand, multi-class classification involves only a single-stage training process, making it more time-efficient.

Sequential classification is particularly useful when the classification task involves a hierarchical structure, where each stage refines the previous stage’s results. This approach helps break down complex classification problems into smaller, manageable tasks, reducing the risk of misclassification and improving overall performance. In contrast, multi-class classification handles all categories simultaneously, which can be more efficient in cases where there is no clear hierarchical relationship between the categories.

### 4.4. Robustness Analysis of the Proposed Method

This study evaluates the robustness of the proposed pork classification and ractopamine detection method under various conditions, including different pork cuts, user-provided additional information, and image blurring due to environmental factors. The sample image quantities used in the experiments for robustness analysis in the three stages are the same as shown in Table 4.

#### 4.4.1. Effect of Different Pork Cuts on the Classification Effectiveness of Pork Origin and Ractopamine Detection

This study proposes using the MobileNet model for pork origin classification and ractopamine detection, enabling consumers to use mobile devices to identify ractopamine in pork. It evaluates the performance of different network models for classifying the five major pork cuts commonly found in retail markets and examines how these cuts affect classification and ractopamine detection.

Figure 17 compares the performance of different network models for pork origin classification across various cuts. Pork shoulder (Meat Cut 1) achieves the highest classification accuracy (83.33%) with MobileNet, due to the relatively uniform distribution of lean meat and fat, making feature differentiation easier during classification. Pork loin (Meat Cut 2) shows lower accuracy (74.44%) because of its uneven lean-to-fat ratio, with a higher proportion of lean meat, making feature differentiation harder. Pork tenderloin (Meat Cut 3) has a lean-to-fat ratio similar to that of pork loin. Also, its classification performance is lower than that of pork shoulder. Pork belly (Meat Cut 4) has more fat than lean meat, and the inconsistent fat-to-lean ratio leads to similar classification challenges as pork loin. Lastly, ham (Meat Cut 5) has a higher lean-to-fat ratio, but it is not as evenly distributed as pork belly, and its features are less discernible, resulting in lower classification accuracy compared to pork shoulder.

Figure 18 compares the performance of different network models for ractopamine detection across various cuts of North American pork. Pork loin and pork belly have uneven lean-to-fat ratios, with pork loin having a higher proportion of lean meat and pork belly containing more fat, making ractopamine detection easier during classification.

#### 4.4.2. Effect of User-Provided Additional Information on Detection Effectiveness

This study focuses on real-time pork classification using images captured on mobile devices while shopping. Providing accurate additional information, such as meat cut type or origin, can improve classification accuracy, while incorrect information may decrease it. This section explores the impact of user-provided additional information on detection effectiveness by simulating scenarios with both correct and incorrect details.

Figure 19 shows that providing additional information improves pork cut classification accuracy, with the CR increasing from 94% to 96% when the correct information is given (a 2% improvement). However, if incorrect information is provided, the CR drops to 78%, representing a 16% decrease. For pork origin classification, the CR drops from 72.67% to 52.67% with incorrect information (a 20% decrease), while correct information increases the CR to 76%, improving by 3.33%. Overall, additional information significantly affects classification accuracy in the sequential classification process.

Without additional information, the MobileNet model achieves 61.27% sequential classification accuracy, while Custom Vision achieves 53.12%. When correct meat cut type information is provided, MobileNet accuracy rises to 63.82% and Custom Vision to 56.92%. If both meat cut type and pork origin information are provided, accuracy improves to 80.67% for MobileNet and 75.33% for Custom Vision. As shown in Table 13, providing correct information in advance significantly enhances the accuracy of detecting ractopamine in pork.

#### 4.4.3. Effect of Image Blur Caused by Environmental Factors on Detection Effectiveness

This study simulates consumers capturing pork images with handheld mobile devices while shopping, accounting for potential issues like label coverage, glare, and image blurriness due to hand movement or moisture. To assess detection performance, the study evaluates images with varying blur levels, including no effect (0), slight (32), moderate (64), and severe (128), as shown in Figure 20.

Figure 21 shows that while blurriness reduces classification accuracy for pork cuts, it remains above 80%. Slightly blurred images perform similarly to clear ones. Due to the large size of the test images (960 × 720), the impact of blurriness on classification is minimal, maintaining high accuracy.

Figure 22 shows that image blurriness decreases the effectiveness of pork origin classification, with accuracy gradually dropping from slight to severe blurriness. The smaller test image size (224 × 224) amplifies the impact of blurriness, making it harder to maintain classification accuracy.

Figure 23 shows that image blurriness decreases all effectiveness indicators in ractopamine detection, with accuracy significantly dropping from slight to severe blurriness. The smaller test images (224 × 224) amplify the impact of blurriness, making it harder to maintain accuracy. Therefore, users should try to avoid blurring as much as possible when capturing images, so as to better maintain the correct classification rate of the images.

### 4.5. Results and Discussion

To critically evaluate the proposed system, we discuss key limitations and considerations related to model robustness, statistical reliability, and real-world applicability. These insights highlight areas for future improvement and underscore the importance of cautious deployment.

While the study does not explicitly analyze resolution-induced bias, the dual-ROI approach reduces potential inconsistencies by tailoring resolutions to task-specific needs, and MobileNet’s performance (96% CR for cuts, 79.11% for origin, 80.67% for ractopamine) suggests robustness across resolutions. However, varying resolutions could introduce differences in feature extraction depth, potentially affecting models like VGG16 more than MobileNet due to parameter complexity. Future work could quantify resolution impacts using a sensitivity analysis to ensure fairness and optimize performance across tasks.

The reported metrics are point estimates derived from the test dataset and do not currently include confidence intervals, standard deviations, or hypothesis testing. To address this limitation transparently, we plan to incorporate these statistical analyses in future work. Specifically, we intend to use bootstrapping to compute confidence intervals and standard deviations for effectiveness metrics and apply hypothesis testing to compare our model’s performance against baseline methods to account for variability and ensure robust evaluations.

While the proposed smartphone-based deep learning system shows good potential for ractopamine screening, its current application is best suited as a preliminary detection tool rather than a final decision-making solution for consumers. Given the lack of regulatory certification, possible algorithmic opacity, and susceptibility to deliberate visual manipulation (e.g., color enhancement or chemical treatments), relying solely on this system for purchase decisions may pose risks. Moreover, heightened public sensitivity surrounding food safety, especially for imports from the United States and Canada, underscores the need for cautious deployment. Therefore, we recommend the system be used to flag suspicious samples for further laboratory confirmation, increasing the efficiency of pork quality monitoring. Future development will focus on improving transparency, robustness, and regulatory compliance to ensure its safe integration into consumer-level applications.

## 5. Concluding Remarks

This study proposes a system for classifying pork cuts and origin, with an extension to detect ractopamine in North American pork. The system allows consumers to use mobile devices in retail settings to analyze pork images and make informed purchasing decisions. The process involves three stages: extracting the outer ROI for meat cut classification, obtaining the inner ROI for pork origin classification, and detecting ractopamine in North American pork. Deep learning models based on MobileNet are evaluated for classification performance and compared to traditional methods and the Custom Vision system on a machine learning cloud platform, selecting the optimal model based on effectiveness and efficiency.

The experimental results show that the MobileNet model outperforms other methods in classification accuracy and F1 score, making it the chosen model for the pork classification and ractopamine detection system. In large-scale tests, the system achieved a 96% classification rate (CR) for pork cuts, 79.11% CR for pork origin classification with a 90.25% F1 score, and 80.67% CR with an 80.56% F1 score for ractopamine detection in North American pork. These results highlight the effectiveness of the proposed method. Future research can extend detection to frozen packaged pork and darkened meat due to their altered visual characteristics (e.g., ice crystal effects or color changes) and develop an app system with cloud technology for data transfer and processing.

## Figures and Tables

**Figure 1 sensors-25-02698-f001:**
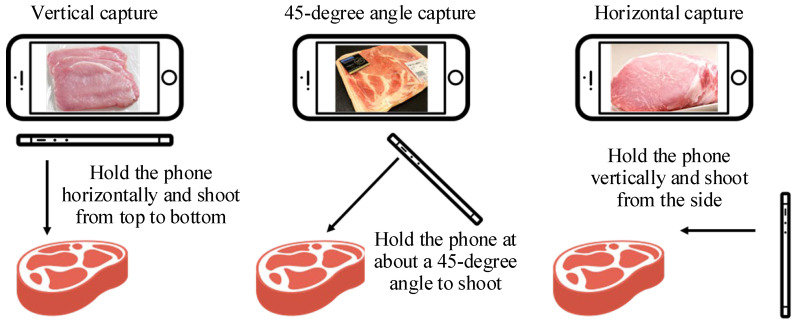
Illustration of three ways to capture pork with a mobile phone.

**Figure 2 sensors-25-02698-f002:**
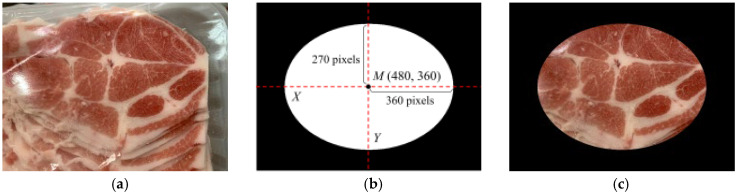
The outer ROI area after adding the mask to the captured image: (**a**) captured image, (**b**) black elliptical mask for outer ROI, (**c**) outer ROI area after adding a mask.

**Figure 3 sensors-25-02698-f003:**
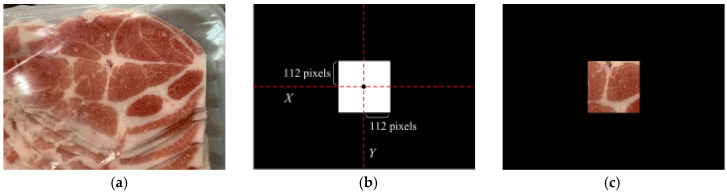
The inner ROI area after adding the mask to the captured image: (**a**) captured image, (**b**) black square mask for inner ROI, (**c**) inner ROI area after adding the mask.

**Figure 4 sensors-25-02698-f004:**
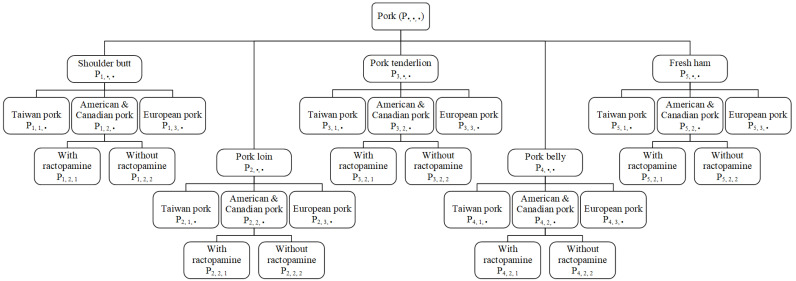
Classification tree diagram of pork classification and ractopamine detection.

**Figure 5 sensors-25-02698-f005:**
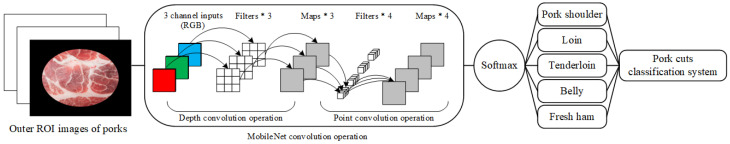
Training procedure for the MobileNet network model for pork cut classification.

**Figure 6 sensors-25-02698-f006:**
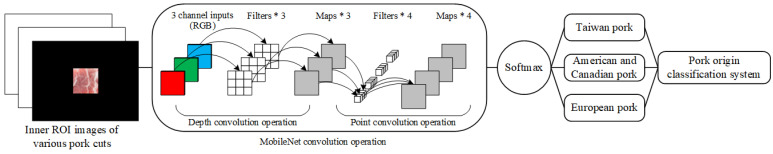
Training procedure for the MobileNet network model for pork origin classification.

**Figure 7 sensors-25-02698-f007:**
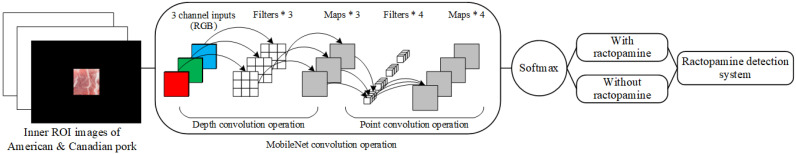
Training process for the MobileNet network model for detecting whether North American pork contains ractopamine.

**Figure 8 sensors-25-02698-f008:**
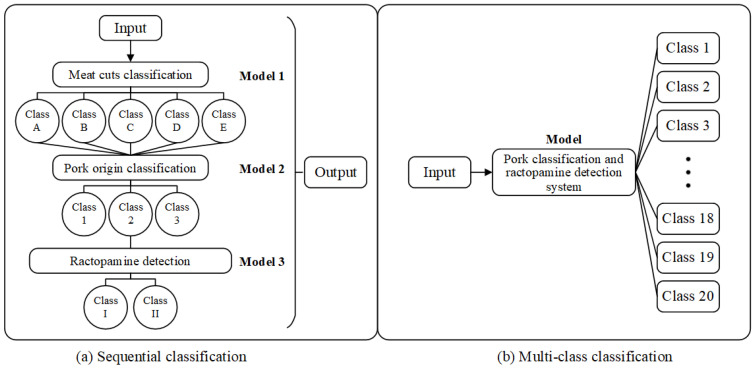
Schematic diagram of overall sequential classification and multi-class classification.

**Figure 9 sensors-25-02698-f009:**
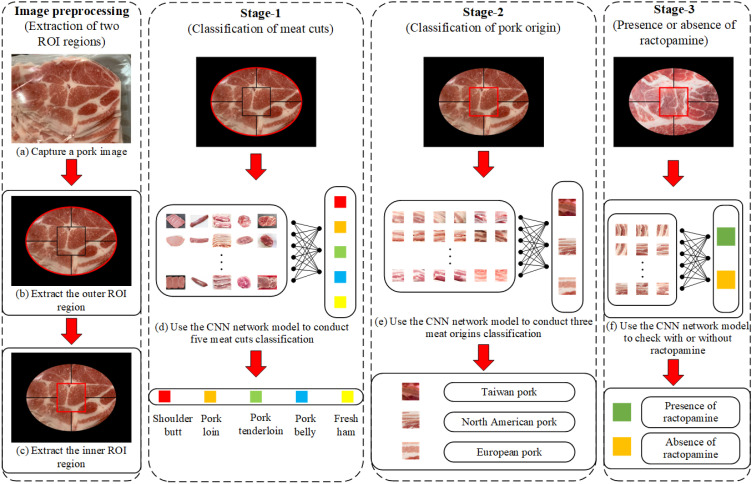
Stage diagram of the pork classification and ractopamine detection system.

**Figure 10 sensors-25-02698-f010:**
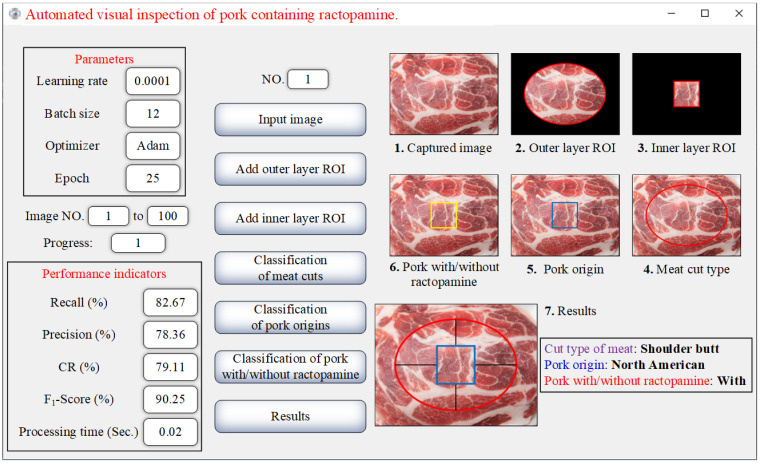
User interface of the detection system developed in this study.

**Figure 11 sensors-25-02698-f011:**
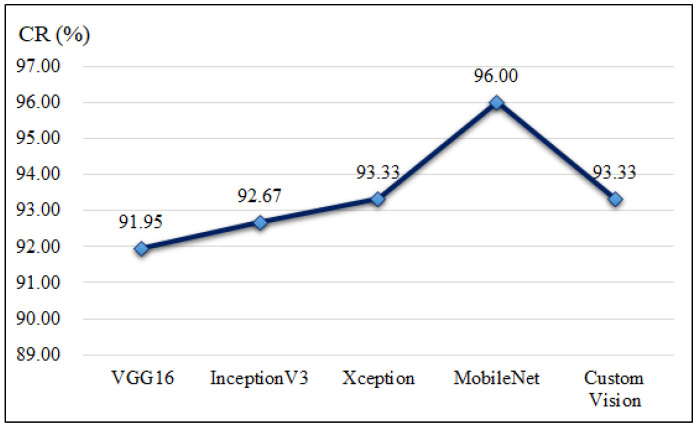
Line chart of the correct classification rates of meat cuts in different deep learning models.

**Figure 12 sensors-25-02698-f012:**
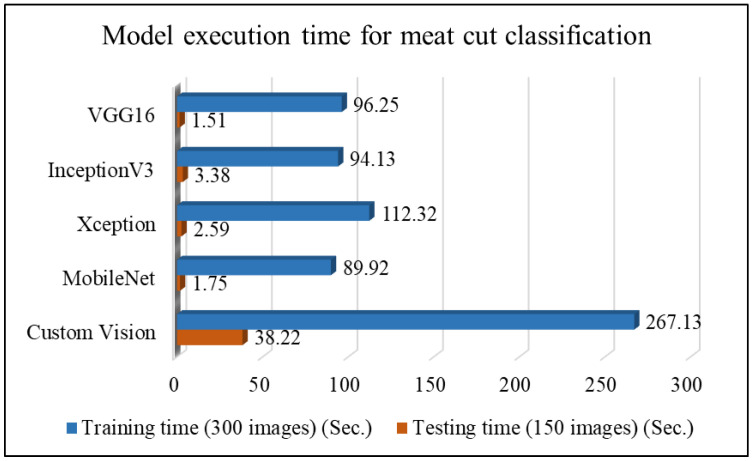
Comparison plot of execution time of meat cut classification using different network models.

**Figure 13 sensors-25-02698-f013:**
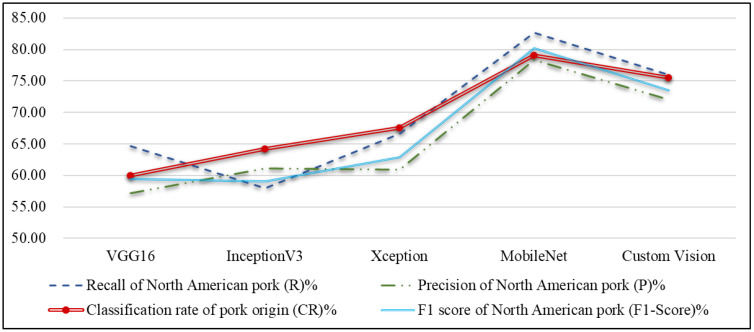
Line chart of effectiveness metrics for pork origin classification using different network models.

**Figure 14 sensors-25-02698-f014:**
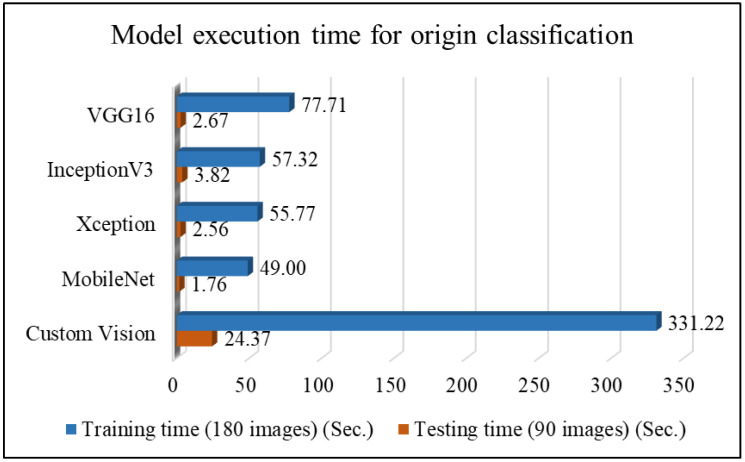
Comparison plot of execution time for pork origin classification using different network models.

**Figure 15 sensors-25-02698-f015:**
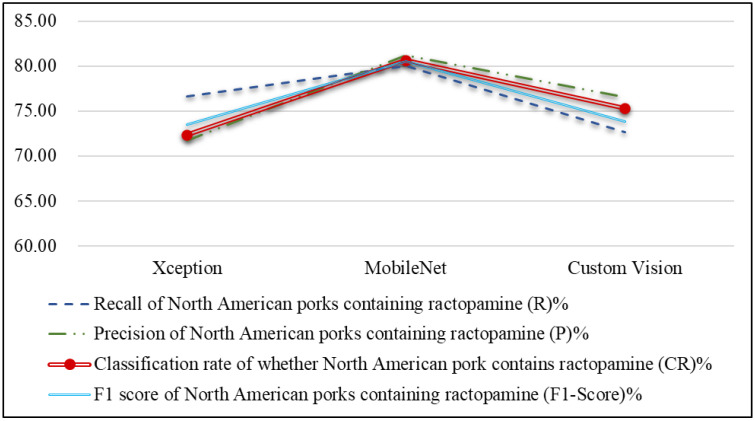
Line chart of effectiveness metrics of ractopamine detection using different network models for North American pork.

**Figure 16 sensors-25-02698-f016:**
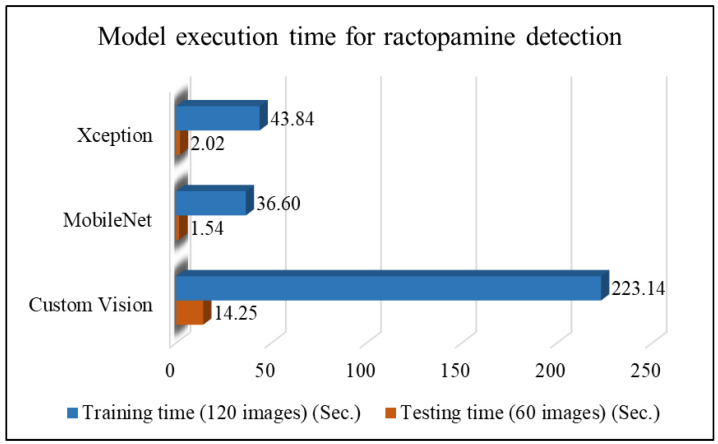
Comparison plot of execution time of different network models for detecting whether North American pork contains ractopamine.

**Figure 17 sensors-25-02698-f017:**
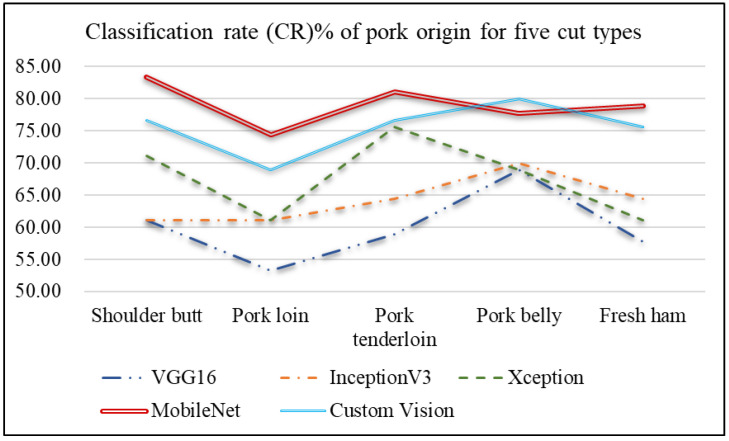
Line chart of the classification rates of pork origin for five meat cut types using different network models.

**Figure 18 sensors-25-02698-f018:**
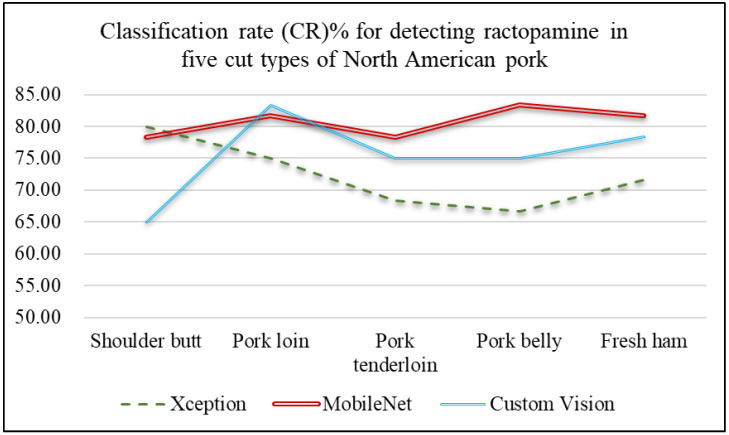
Line chart of the classification rates of ractopamine detection for five meat cut types of North American pork using different network models.

**Figure 19 sensors-25-02698-f019:**
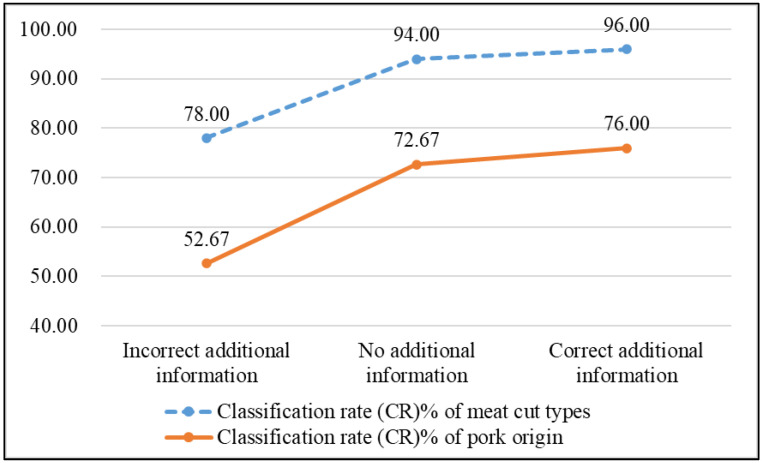
Line chart of classification rates for meat cuts and pork origin when users provide additional information.

**Figure 20 sensors-25-02698-f020:**
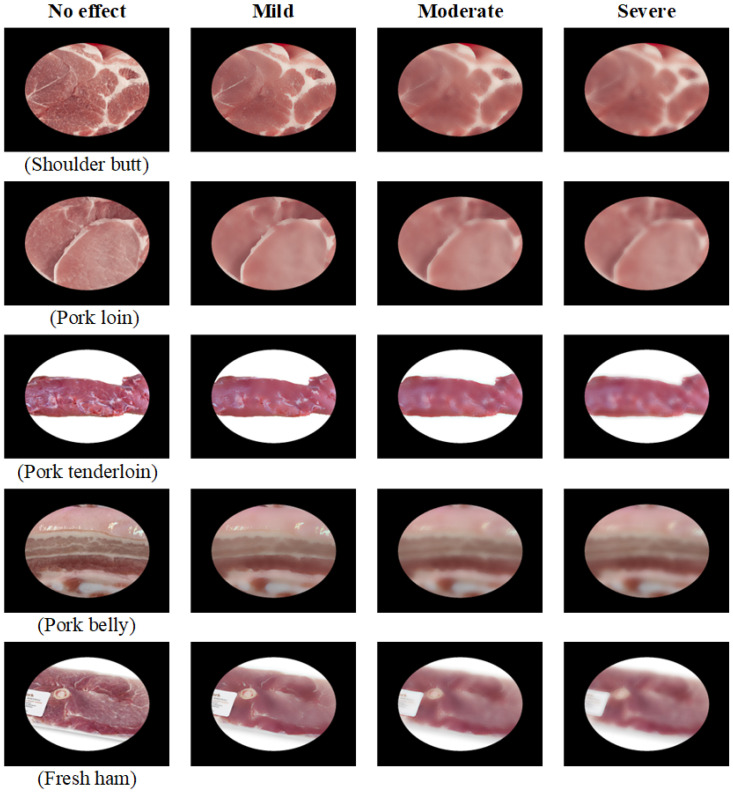
Comparison of images with different levels of blurriness for five meat cut types.

**Figure 21 sensors-25-02698-f021:**
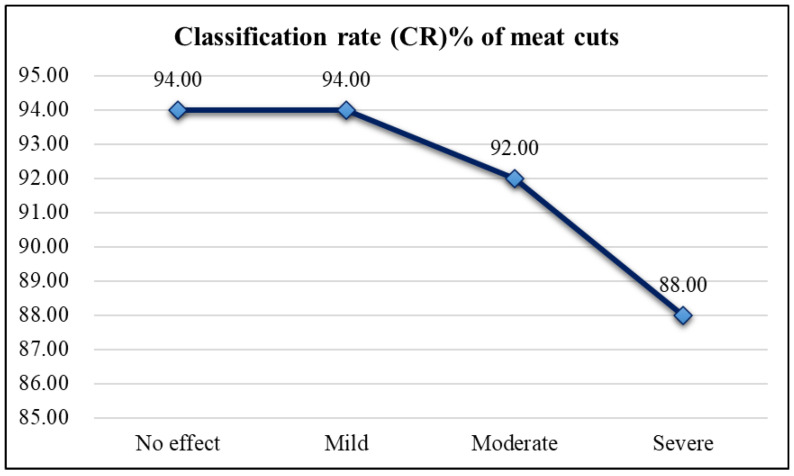
Line chart of classification rates of meat cuts for images with different levels of blurriness.

**Figure 22 sensors-25-02698-f022:**
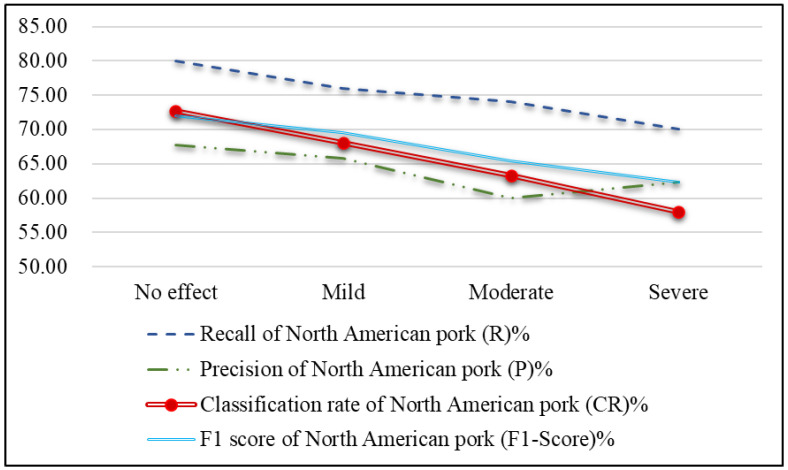
Line chart of effectiveness metrics of pork origin for images with different levels of blurriness.

**Figure 23 sensors-25-02698-f023:**
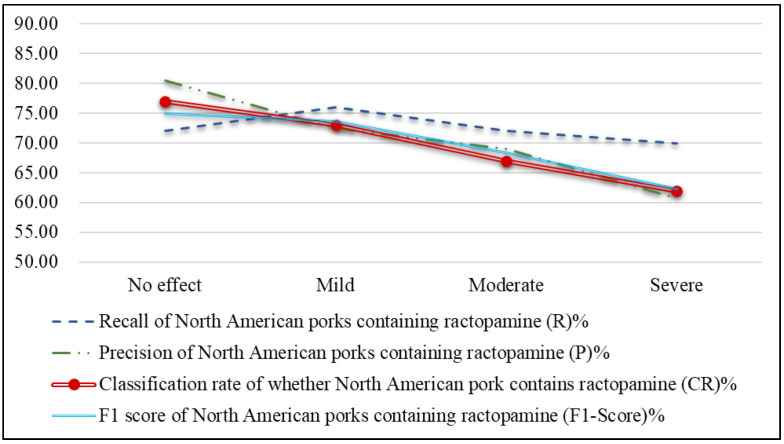
Line chart of effectiveness metrics of ractopamine detection for images with different levels of blurriness.

**Table 1 sensors-25-02698-t001:** A comparison of the appearance characteristics of representative meat cuts.

Main Pork Cuts	Representative Meat Cuts	Appearance Characteristics (Shape, Color, Texture)
Shoulder	Shoulder butt 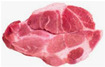	The pork shoulder butt is an irregular and bulky shape with some uneven edges. It is dark pink to red in color, featuring visible marbling and connective tissue. The texture is coarse-grained and dense due to the presence of muscle fibers and intramuscular fat.
Loin	Pork loin 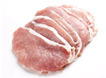	The pork loin is long and rectangular when whole, while pork chops are round or oval-shaped. It has a light pink color with a small amount of marbling and an outer fat cap. The texture is fine-grained and smooth, feeling relatively firm when raw.
Tenderloin	Pork tenderloin 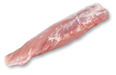	The pork tenderloin is long, narrow, and cylindrical with tapered ends. It has a pale pink color with very little visible fat. The texture is extremely smooth and soft, almost velvety, with a thin outer membrane known as silverskin.
Belly	Pork belly 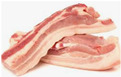	The pork belly is rectangular with alternating layers of fat and lean meat, while spare ribs have an elongated shape with exposed rib bones. The color ranges from light pink to reddish with distinct white fat layers. The texture is soft and pliable for pork belly, whereas spare ribs have a firm but slightly flexible structure due to the bones.
Leg	Fresh ham 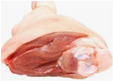	The fresh ham is large, oval, and compact in shape, often appearing rounded. It is pale pink, while cured ham is a deeper red with a slightly glossy surface. The texture is firm and dense, with a smooth surface and a thick fat cap, while cured ham may have a slightly dry or glossy exterior.

**Table 2 sensors-25-02698-t002:** Comparison of visual characteristics between Taiwan pork and U.S. ractopamine-fed pork.

Name	Taiwan Pork	U.S. Ractopamine-Fed Pork
Pork image	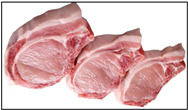	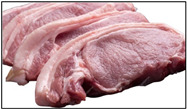
Surface Characteristics	1. The lean meat is light pink with a firm texture. 2. A distinct fat layer is present between the skin and lean meat, with a subcutaneous fat thickness of at least 1–2 cm or more. 3. The meat is relatively firm, and slices (2–3 fingers wide) can stand upright. 4. Fat and lean meat gradually transition without a distinct boundary, and no visible moisture exudation (sweating) occurs. 5. The fat is pure white, with clear marbling in the muscle, and the muscle surface appears smooth.	1. The lean meat is darker and bright red, with a more vivid coloration and loosely textured fibers. 2. The fat and lean meat have a clear separation, with a thinner fat layer (typically <1 cm), sometimes with the skin attached directly to the lean meat. 3. The meat is softer, and slices (2–3 fingers wide) cannot stand upright. 4. Moisture exudation (sweating) may be observed between the lean meat and fat, making the fat especially thin. 5. The muscle on the hindquarters appears plumper and more protruding, with thinner fat that is less white in color. 6. The fat layer in the groin area on both sides shows a denser capillary network, sometimes appearing congested.

**Table 3 sensors-25-02698-t003:** Parameter settings and default values of network models for different image sizes in this study.

Parameters of Network Models	Classification of Meat Cuts	Classification of Pork Origins and Detection of Ractopamine
Image size	960 × 720	224 × 224
Learning rate	0.0001	0.0001
Batch size	16	16
Optimizer	Adam	Adam
Number of epochs	20	20

**Table 4 sensors-25-02698-t004:** Sample image quantities used in small-sample experiments for parameter setting.

Small-Sample Experiments	Images for Meat Cuts (5 Meat Cuts)		Images for Pork Origin (3 Pork Origins)		Images for Ractopamine (2 Outcomes)	
	Training (5 × 20)	Testing (5 × 10)	Training (5 × 3 × 20)	Testing (5 × 3 × 10)	Training (5 × 2 × 20)	Testing (5 × 2 × 10)
Image number	100	50	300	150	200	100
Total	150		450		300	
Image size	960 × 720		224 × 224		224 × 224	

**Table 5 sensors-25-02698-t005:** The optimized parameter settings for the network models in meat cut classification, pork origin classification, and ractopamine detection.

Model Parameters	Meat Cut Classification	Pork Origin Classification	Ractopamine Detection
Image size	960 × 720	224 × 224	224 × 224
Learning rate	0.0001	0.0001	0.0001
Batch size	12	12	12
Optimizer	Adam	Adam	Adam
Number of epochs	25	25	25

**Table 6 sensors-25-02698-t006:** Sample image quantities used in large-sample experiments for performance evaluation.

Large-Sample Experiments	Images for Meat Cuts (5 Meat Cuts)		Images For Pork Origin (3 Pork Origins)		Images for Ractopamine (2 Outcomes)	
	Training (5 × 60)	Testing (5 × 30)	Training (5 × 3 × 60)	Testing (5 × 3 × 30)	Training (5 × 2 × 60)	Testing (5 × 2 × 30)
Image number	300	150	900	450	600	300
Total	450		1350		900	
Image size	960 × 720		224 × 224		224 × 224	

**Table 7 sensors-25-02698-t007:** Comparison of model execution time for meat cut classification using different network models.

Network Model	VGG16	InceptionV3	Xception	MobileNet	Custom Vision
Training time (300 images) (s)	96.25	94.13	112.32	89.92	267.13
Testing time (150 images) (s)	1.51	3.38	2.59	1.75	38.22
Testing time/image (s)	0.010	0.023	0.017	0.012	0.255

**Table 8 sensors-25-02698-t008:** Comparison of the effectiveness metrics of different network models for the origin classification of pork cuts.

Effectiveness Metrics	VGG16	InceptionV3	Xception	MobileNet	Custom Vision
Recall of North American pork (R)%	64.67	58.00	66.67	82.67	76.00
Precision of North American pork (P)%	57.12	61.12	60.94	78.36	72.00
F1 score of North American pork (F1 Score)%	59.43	59.06	62.89	90.25	73.51
Classification rate of pork origin (CR)%	60.00	64.22	67.56	79.11	75.56

**Table 9 sensors-25-02698-t009:** Comparison of model execution time for pork origin classification using different network models.

Network Model	VGG16	InceptionV3	Xception	MobileNet	Custom Vision
Training time (180 images) (s)	77.71	57.32	55.77	49.00	331.22
Testing time (90 images) (s)	2.67	3.82	2.56	1.76	24.37
Testing time/image (s)	0.030	0.042	0.028	0.020	0.271

**Table 10 sensors-25-02698-t010:** Comparison of the effectiveness metrics of ractopamine detection using different network models for North American pork.

Effectiveness Metrics	Xception	MobileNet	Custom Vision
Recall of North American pork containing ractopamine (R)%	76.67	80.00	72.67
Precision of North American pork containing ractopamine (P)%	71.75	81.26	76.60
F1 score of North American pork containing ractopamine (F1 Score)%	73.54	80.56	73.83
Classification rate of whether North American pork contains ractopamine (CR)%	72.33	80.67	75.33

**Table 11 sensors-25-02698-t011:** Comparison of model execution time of different network models for detecting whether North American pork contains ractopamine.

Network Model	Xception	MobileNet	Custom Vision
Training time (120 images) (s)	43.84	36.60	223.14
Testing time (60 images) (s)	2.02	1.54	14.25
Testing time/image (s)	0.034	0.026	0.238

**Table 12 sensors-25-02698-t012:** Comparison table of performance metrics for various classification methods in the pork classification and ractopamine detection system.

Performance Metrics	MobileNet		Custom Vision	
	CR (%)	Time	CR (%)	Time
Classification rate (CR)% and testing time (s) of meat cuts	96.00	0.012	93.33	0.255
Classification rate (CR)% and testing time (s) of pork origin	79.11	0.020	75.56	0.271
Classification rate (CR)% and testing time (s) of ractopamine detection	80.67	0.026	75.33	0.238
Classification rate (CR)% and testing time (s) of overall sequential classification	61.27	0.058	53.12	0.764
Classification rate (CR)% and testing time (s) of overall multi-class classification	60.06	0.046	61.80	0.582

**Table 13 sensors-25-02698-t013:** Comparison of classification rates of sequential classification methods when users provide different degrees of correct additional information.

Learning Models	No Additional Information	Correct Meat Cut Information Provided	Correct Meat Cut and Pork Origin Information Provided
MobileNet CR (%)	61.27	63.82	80.67
Custom Vision CR (%)	53.12	56.92	75.33

## Data Availability

The data will be made available on request.

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
