# Peer review of "Smartphone-Based Deep Learning System for Detecting Ractopamine-Fed Pork Using Visual Classification Techniques"

_sensors, 2025, doi:10.3390/s25092698_

Round 1

Reviewer 1 Report

Comments and Suggestions for Authors

The manuscript entitled "Smartphone-based Deep Learning System for Detecting Ractopamine-fed Pork Using Visual Classification Techniques" proposes a three-stage sequential classification method to detecting ractopamine-fed Pork based on visual classification techniques. The concept seems to be interesting and it's very practical. The process and results of the study are very detailed, but the language is too verbose, the contents with the same meaning is explained repeatedly. Such as :

1) Line 101-104 and Line 253-256, repeatly

2) part 3.3.1 and part 2.4, repeatly

3) “the three-stage sequential classification method” was explained several times

4) the advantages of “the three-stage sequential classification method” are explained many times

5)Line 440-442 repeatly

6) image size: 960â…¹700(meat cut), 224â…¹224(pork origin and ractopamine) repeated several times

……

the language is too verbose and needs to be carefully condensed.

Comments on the Quality of English Language

The manuscript entitled "Smartphone-based Deep Learning System for Detecting Ractopamine-fed Pork Using Visual Classification Techniques" proposes a three-stage sequential classification method to detecting ractopamine-fed Pork based on visual classification techniques. The concept seems to be interesting and it's very practical. The process and results of the study are very detailed, but the language is too verbose, the contents with the same meaning is explained repeatedly. Such as :

1) Line 101-104 and Line 253-256, repeatly

2) part 3.3.1 and part 2.4, repeatly

3) “the three-stage sequential classification method” was explained several times

4) the advantages of “the three-stage sequential classification method” are explained many times

5)Line 440-442 repeatly

6) image size: 960â…¹700(meat cut), 224â…¹224(pork origin and ractopamine) repeated several times

……

the language is too verbose and needs to be carefully condensed.

Reviewer 2 Report

Comments and Suggestions for Authors

This study proposes a smartphone-based system with cloud transmission to help consumers identify residual ractopamine in North American pork. The Subject of the paper is interesting and helpful for stakeholders. However, the content of the paper needs deep revision and some sections, such as the introduction and related work, should be merged. The aim of the study has been mentioned several times in different sections of the paper. In fact, the author should highlight the current gap due to the subject and underscore how this study addresses the problems. In addition, a lack of enough discussion in the results and discussion section is tangible. The authors should compare their results with the literature and describe the superiority of their work with the recently published papers. After addressing the mentioned and following comments, the paper can be appropriate for publication.

L18 and L21- specify the abbreviations of ROI (region of interest) and CR in the text.

L63- Illustration of five major pork cuts and representative meat cuts is not essential, please remove it.

L31- The correlation between ractopamine and meat redness should be backed with a reference.

L66- In Table 2, these visual differences can be highly subjective and lighting-dependent. So, mention whether lighting conditions were standardized in the dataset.

L133- the authors mentioned ‘’However, none of the reviewed studies mention the use of computer vision technology to detect whether pork has been produced using ractopamine’’. In fact, I doubt this claim. For example, you can find similar research in the: Yin, T., Peng, Y., Li, Y., Chao, K., Nie, S., Tao, F., & Zuo, J. (2025). A multilevel cooperative attention network of precise quantitative analysis for predicting ractopamine concentration via adaptive weighted feature selection and multichannel feature fusion. Food Chemistry, 464, 141884.

L 163- The authors mentioned ‘’. However, this study explores the use of 162 image processing combined with deep learning models to classify pork and further deter-163 mine whether it contains ractopamine.’’ I recommend revising the introduction and related work section deeply and mentioning the aims of this study in one paragraph, not separately.

L180-248 It seems that just a sentence of abstract from each published paper was mentioned. So, it can not be helpful and interesting for the readers. The author should highlight the current gap and underscore their proposed method

L186-198. The criteria for choosing these models is missing, so add its justification.

L231-248 Clarify how much of the model was customized vs. pre-trained. Was data augmentation used?

L249- Several times the aim of study is mentioned.

L452- 599 here is no mention of confidence intervals, standard deviations, or hypothesis testing to support the reported classification rates or F1 scores.  The results report the performance of MobileNet and other deep learning models, but there is no comparison against classical machine learning methods like SVM.

L456- Image resolutions vary between classification tasks (960×720 for pork cuts and 224×224 for others), but there’s no discussion of how this might affect the network's performance or introduce bias.

L492 and 517- in table 4 and 6, the method for selecting training and testing images is not clearly explained, particularly regarding class balancing.

L523- There’s no validation or cross-checking of ground-truth labels

Reviewer 3 Report

Comments and Suggestions for Authors

The work develops quite promising sensor technology for meat quality control, such as smartphone-based deep learning systems for detecting pork tainted with ractopamine. Indeed, training an expert to evaluate the appearance of meat is long and expensive, since the quality of meat depends on a large number of factors. Therefore, replacing an expert with an application that can identify a suspicious product is very profitable. The idea of machine learning and recognizing the quality of pork and meat is relevant and has not yet been resolved. It is unlikely that this approach can be recommended for consumer assessment, due to the possibility of manipulating consumer preferences by the developer in the absence of transparency of the evaluation parameters. The algorithm must be certified though. With high concern among the population, it is easier to expect a refusal to buy American and Canadian pork, and a reduction in meat consumption in general. In addition, the use of the application can give rise to a new round of falsification in order to deceive the application and get into its algorithm with the help of color fixers, chemical brighteners, drawing, etc. At the same time, the opinion made by the application on the quality can be considered by the consumer as more significant than his own, which can potentially lead to poisoning. I think it is more relevant to reduce the number of meat samples examined using laboratory methods by selecting only suspicious samples using an application, which can increase the efficiency of detecting adulteration. Despite the problematic aspects I have mentioned above, the work is worth publishing after addressing the following comments:
1. The article does not indicate whether the proposed method can be used to distinguish meat with ractopamine from defrosted meat or meat darkened due to storage.
2. What is the percentage of error in the determination in the case of natural darkening of meat?
3. Is the method generally suitable for defrosted meat?

Round 2

Reviewer 2 Report

Comments and Suggestions for Authors

All of the comments were considered and addressed. The current version of the paper is appropriate for publication.